# Species-Level Versus Community-Level Responses to Microhabitat Type and Diversity in an Experimental Plant Community

**Bing Hu** [1,2]**, Zhu-Wen Xu** [1,]*** , Wei Xue** [2,]*** and Fei-Hai Yu** [2]

1   Key Lab Grassland Ecology, School of Ecology & Environment, Inner Mongolia University, Hohhot 010021, China
2   Institute of Wetland Ecology & Clone Ecology/Zhejiang Provincial Key Laboratory of Evolutionary Ecology and Conservation, Taizhou University, Taizhou 318000, China
*   Correspondence: zwxu@imu.edu.cn (Z.-W.X.); x_wei1988@163.com (W.X.)

**Abstract:** Soil heterogeneity has been theoretically predicted to promote plant performance, but empirical evidence is often mixed. By focusing on a gradient of microhabitats (single, three and six types of microhabitats), we tested the hypothesis that increasing the number of microhabitats promotes the diversity and productivity of plant communities. We created different types of microhabitats by adding different forms of nitrogen ($NH_4^+$, $NO_3^-$ and glycine) or combinations of these nitrogen in an even or patchy way. Community biomass, but not evenness, differed in different types of single-microhabitat. Increasing the number of microhabitat types did not alter community growth or evenness, but it increased the difference in the relative abundance of plant species within the community. These results suggest that microhabitat diversity can influence plant competitive hierarchies, and that this effect can lead to changed community growth, but may not be decisive for plant evenness.

**Keywords:** community evenness; productivity; multi-microhabitat; nitrogen forms; plant diversity; soil heterogeneity





## 1. Introduction

Soil heterogeneity is recognized as one of the complex mechanisms maintaining plant species diversity [1]. The 'environmental heterogeneity' hypothesis holds that heterogeneous environments can sustain more species than homogeneous environments, because a heterogeneous environment provides more niches, and thus allows more species to coexist [2–5]. Experimental studies on the heterogeneity–diversity relationship (HDR) have yielded different results. Positive HDRs were confirmed in several experimental results [6–8], but negative [9–11], neutral [12] and unimodal HDRs [13] have also been reported. However, these studies generally considered only a limited number of microhabitat types, consisting of low- and high-nutrient soil patches; we know little about how increasing the number of microhabitat types may modify the heterogeneity effects [14].

Soils in nature often contain a wider range of microhabitat types, and thus can support more species, due to more available niches [15–18]; therefore, we expect that increasing the number of microhabitat types contributes to the coexistence of more species, and consequently to a more diverse community. In a given area, however, increasing the number of microhabitat types would result in a decrease of the area for each microhabitat type [18–20]; therefore, increasing the number of microhabitat types may also lead to the exclusion of plant species, and consequently to a more homogenized community. In a previous study, Xue, Huang and Yu [14] found that increasing the number of soil-patch types characterized by different nutrient levels can promote plant community evenness by reducing competitive-ability difference between plant species within the community; however, this

study failed to consider the possible confounding effect of the spatial distribution of different forms of nitrogen. We know little about how different forms of nitrogen and their spatial distribution may influence plant community responses.

Plants differ greatly in their requirements and abilities, in utilizing different forms of nitrogen [21–26]; therefore, a plant community will be dominated by the species exhibiting the phenotypes best-fitted to absorb the dominant form of nitrogen. Importantly, the existence of different forms of nitrogen and their spatial distributions may provide the opportunity for the coexistence of plant species with differing nitrogen-use strategies [27]. This hypothesis may provide an alternative explanation for the coexistence of plant species that largely overlap in resource utilization [28–30]; however, we lack the evidence of how different forms of nitrogen and their spatial distributions may influence plant community responses.

For this study, in a greenhouse experiment, we constructed plant communities in soils consisting of single, three and six types of microhabitats. The different microhabitats were created by adding different forms of nitrogen ($NH_4^+$, $NO_3^-$ and glycine), or combinations of these forms of nitrogen, in an even or patchy way in the soil. We mainly tested two hypotheses: (1) plant communities are dominated by different plant species, depending on the type of the single-microhabitats, which results in different growth and evenness in plant communities; (2) increasing the number of microhabitat types increases the growth and evenness of plant communities, due to the more equal growth of the plant species within the communities.

## 2. Materials and Methods

### 2.1. The Species

We purchased seeds of six species (two forbs, i.e., *Taraxacum mongolicum* Hand.-Mazz. and *Plantago depressa* L.; two grasses, i.e., *Elymus dahuricus* Turcz. and *Lolium perenne* L.; and two legumes, i.e., *Medicago sativa* L. and *Trifolium repens* L.) from a local supplier. The seeds were kept at 4 °C before germination. On 8 October 2021, we sowed the seeds of the six species separately in trays filled with peat soil. The trays were placed in a greenhouse at Taizhou University in Taizhou, Zhejiang Province, China. Water was supplied to the trays each day, to keep the soil moist to promote germination. After two weeks, similar-sized seedlings were selected and used in the experiment described below.

### 2.2. The Experiment

On 18 October 2021, we constructed experimental plant communities in 35 pots (each 32 cm long × 22.5 cm wide × 9 cm high) filled with a 1:1 (v:v) mixture of potting soil (Hebei dewoduo Fertilizer Co., Ltd., Hengshui, China) and river sand. To construct the communities, we first divided each pot into 12 patches (each 8.0 cm long × 7.5 cm wide), and then transplanted 4 seedlings of each of the six species (24 seedlings in total) into each pot. The 24 seedlings were randomly assigned to the 12 patches, and each patch was grown with 2 seedlings. No barriers were set up between the adjacent patches, so that the plants could grow across the patches. Dead seedlings were replaced during the first two weeks of the experiment.

Ten days after transplantation, on 28 October 2021, the 35 communities (in the 35 pots) were randomly assigned to seven single-microhabitat treatments and two multi-microhabitat treatments (three- and six-microhabitat treatments). We first created seven types of N solutions with different N forms: (1) ammonium-N only; (2) nitrate-N only; (3) organic-N only; (4) an equal mixture of ammonium-N and nitrate-N; (5) an equal mixture of ammonium-N and organic-N; (6) an equal mixture of nitrate-N and organic-N; and (7) an equal mixture of ammonium-N, nitrate-N and organic-N. Ammonium-N was provided by $NH_4Cl$, nitrite-N by $Ca(NO_3)_2$, and organic-N by glycine. To create the seven N solutions, we dissolved 13.75 g of $NH_4Cl$, 30.36 g of $Ca(NO_3)_2 \cdot 4H_2O$, 19.30 g of glycine, a mixture of 6.88 g of $NH_4Cl$ and 15.18 g of $Ca(NO_3)_2 \cdot 4H_2O$, a mixture of 6.88 g of $NH_4Cl$ and 9.65 g of glycine, a mixture of 15.18 g of $Ca(NO_3)_2 \cdot 4H_2O$ and 9.65 g of glycine mixture, and a

mixture of 4.58 g of $NH_4Cl$, 10.12 g of $Ca(NO_3)_2 \cdot 4H_2O$ and 6.43 g of glycine separately into 600 mL water. In this way, the total amount of N in the seven types of solutions was equal.

For the single-microhabitat treatment, we added 10 mL of a solution into each of 12 patches within a pot, and thus a total of 120 mL of the solution (equivalent to 10 g $N/m^2$) was added into the pot. Each of the seven solutions were applied separately into 3 pots, resulting in 21 pots. For the three-microhabitat treatment, we randomly selected seven combinations of three solutions from the seven solutions described above, such that each solution occurred at the same number of times (each solution was present three times in the selected combinations in this study). The seven combinations were treated as seven replicates for the three-microhabitat treatment. For a given combination, we added 10 mL of each of the three component solutions into four randomly selected patches within a pot, so that the pot consisted of three different microhabitats each with four patches (Figure 1). For the six-microhabitat treatment, all the seven combinations of six solutions were selected and used as replicates. For a given combination, we added 10 mL of each of the six component solutions into two randomly selected patches within a pot, so that the pot consisted of six different microhabitats, each with two patches.

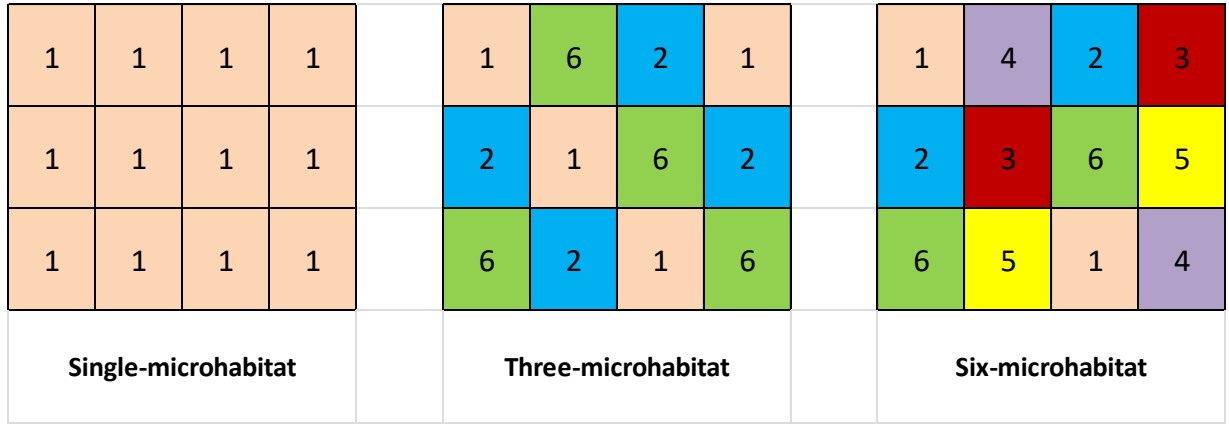

**Figure 1.** Schematic representation of the experimental design. There are seven single microhabitat treatments and two multi-microhabitat treatments (three-microhabitat and six-microhabitat) in the experiment. The numbers (1–6) and their corresponding color represent different single-microhabitat treatments, respectively. The figure above shows only one possible combination of the experimental design.

The solution was added to the corresponding pots once every two weeks, until the termination of the experiment on 10 March 2022. During the experiment, the daily mean temperature and relative humidity in the greenhouse were 23.5 °C and 72.5%, respectively. Water was added to each pot every other day, to keep the soil moist.

*2.3. Harvest and Measurement*

At the end of the experiment, we collected the aboveground parts of each plant in each pot, and the plants were sorted by species. After that, we harvested the roots of all the species together in each pot. All plant materials were dried in an oven (70 °C) for 48 h, and weighed to obtain dry biomass.

*2.4. Data Analysis*

We calculated evenness for each community (pot) as: $J = \frac{-\sum_{i=1}^{s} P_i ln P_i}{ln S}$, where $S$ was the total number of species in the community, and $P_i$ was the aboveground biomass of species $i$ divided by the total aboveground biomass of the community. We used one-way ANOVA to examine the effect of different types of single-microhabitat treatments on shoot biomass, root biomass and evenness of the plant community. In this analysis, only data collected from single-microhabitat treatments were used. We also used one-way ANOVA

to examine the effect of microhabitat diversity (single-microhabitat vs. three-microhabitat vs. six-microhabitat) on shoot biomass, root biomass and evenness of the plant community. In this analysis, data in all the treatments were used. In both the data analyses, when a significant effect was detected, a post-hoc *Duncan's* test was used, to compare the means of different treatments.

To directly examine how different types of single-microhabitat/microhabitat diversity may influence the competitive hierarchies of the component plant species in a community, we used a linear mixed-effect model. In the model, the relative abundance of each of the component species in the community was used as a responsible variable, and the type of single-microhabitat/microhabitat diversity was used as an explanatory variable. We also included pot identity as a random factor in the model, to account for the non-independence of the component species within a community. When a significant effect was detected, a post-hoc *Duncan's* test was used to compare means between different treatments. The shoot biomass of each species was analyzed in the same way.

To explore the possible underlying mechanisms, we calculated the competitive ability (*CA*) of each plant species in a community, relative to a perfectly even community (i.e., the relative abundance of each of the six plant species was one sixth) as: $CA_i = log \frac{RA_i}{1/6}$, where $CA_i$ was the competitive ability of species $i$ in a community, and $RA_i$ was the relative abundance of species $i$ in the community [14]. We then calculated the difference in the competitive ability between species within the community, by subtracting the lowest *CA* from the highest *CA*. The relationships between the difference in competitive ability and shoot biomass, root biomass and evenness were analyzed, using separate linear regression. All analyses were performed with IBM SPSS statistics 22. Where necessary, data were log-transformed to satisfy homogeneity of variance.

## 3. Results

Both shoot and root biomass varied in response to different types of N solutions (Figure 2A,B). In general, the shoot biomass and root biomass of the community treated with organic-N were the highest, while those of the community treated with nitrate-N were the lowest. Neither shoot biomass nor root biomass differed significantly among the communities treated with a mixture of nitrite- and ammonium-N, a mixture of ammonium-N and organic-N and a mixture of all three N forms (Figure 2A,B). By contrast, plant community evenness did not differ among the seven types of the single-microhabitat treatments (Figure 2C). Increasing the number of microhabitats from one to six did not alter shoot biomass, root biomass or evenness of the community (Figure 2D,F).

The shoot biomass of the community was significantly positivity correlated to difference in the competitive ability of the species within the community (Figure 3A). A reversed pattern was found for community evenness (Figure 3C). However, there was no significant relationship between the root biomass of the community and difference in the competitive ability of the species within the community (Figure 3B).

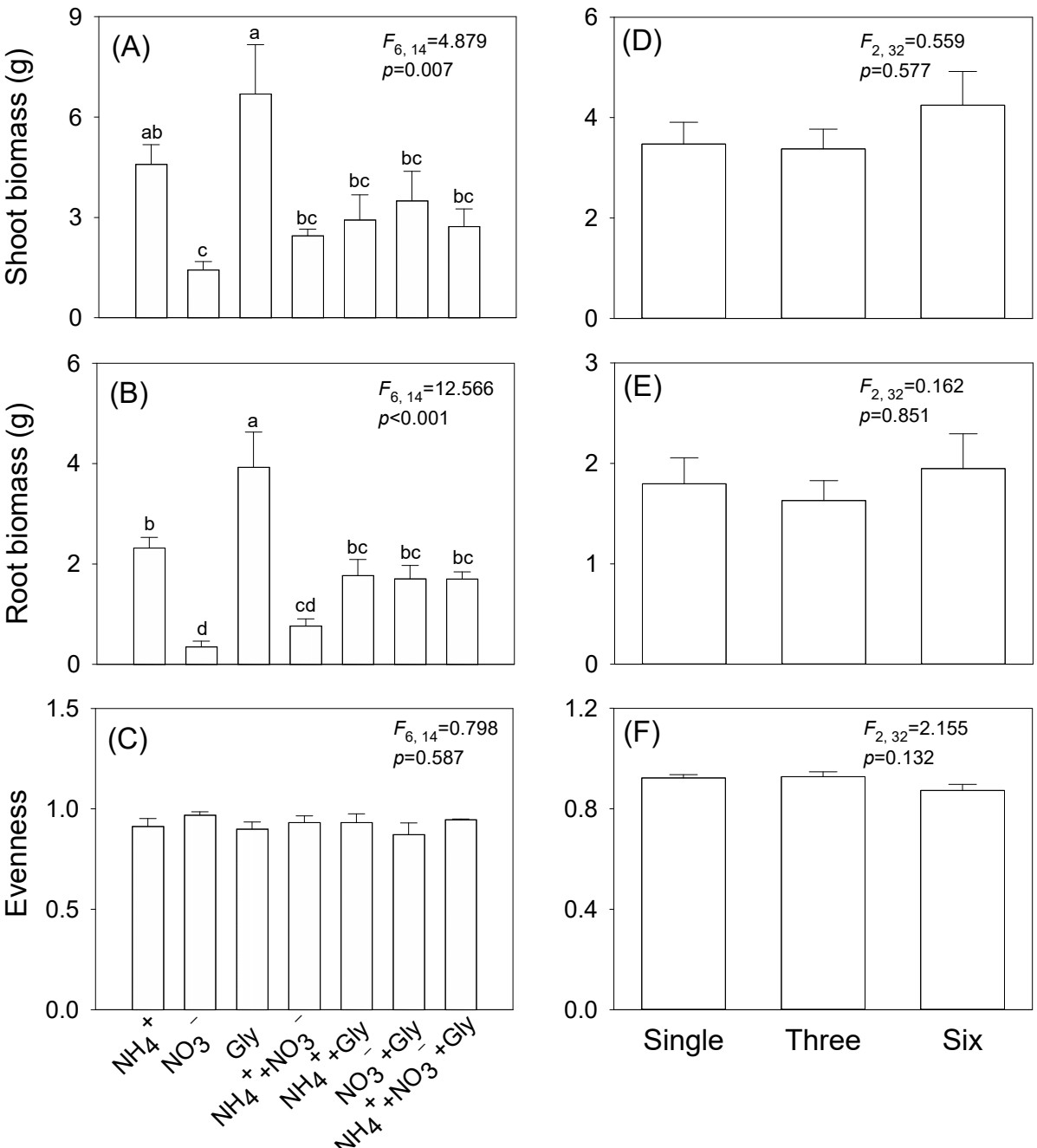

**Figure 2.** Shoot biomass (**A**,**D**), root biomass (**B**,**E**), and evenness (**C**,**F**) in response to different types of single-microhabitats (**A**–**C**) and increasing number of microhabitats (**D**–**F**). *F*- and *p*-values based on one-way ANOVA are given. 'Single', 'Three' and 'Six' represent single-microhabitat and multi-microhabitat with three and six microhabitats, respectively. Different letters (a–d) at the end of bars indicate significant difference in shoot biomass or root biomass among the communities under each treatment.

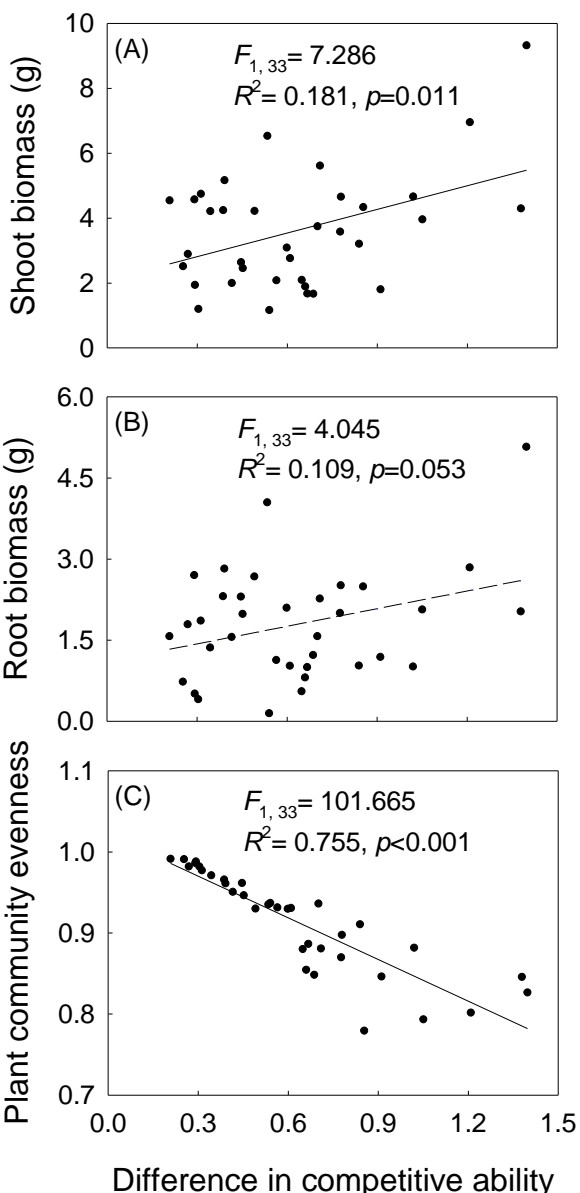

**Figure 3.** Relationship between difference in competitive ability and shoot biomass (**A**), root biomass (**B**) and evenness (**C**) of the community. *F*-, $R^2$ and *p*-values based linear regressions are also given. Solid lines represent regression lines.

The relative abundance of different species varied in response to different types of single-microhabitat (Figure 4A). *Trifolium repens* and *Taraxacum mongolicum* were the dominant species in the community treated with nitrite-N and organic-N, respectively (Figure 4A). *Medicago sativa* was the dominant species in the community treated with a mixture of ammonium-N and organic-N, a mixture of nitrate-N and organic-N and a mixture of all three N forms. However, the relative abundance did not differ among the six component species when the community was treated with ammonium-N and a mixture of ammonium- and nitrate-N (Figure 4A). The relative abundance of different species also varied in response to increasing the number of solution types (Figure 4B). In general, *Medicago sativa* was the most dominant species in both the single- and multi-microhabitat treatments, but its dominance was much more pronounced in the six-microhabitat treatment than in the single- and three-microhabitat treatments (Figure 4B). The shoot biomass of the plant species within the community showed a similar pattern in response to different types of single-microhabitat and microhabitat diversity (Figure 5).

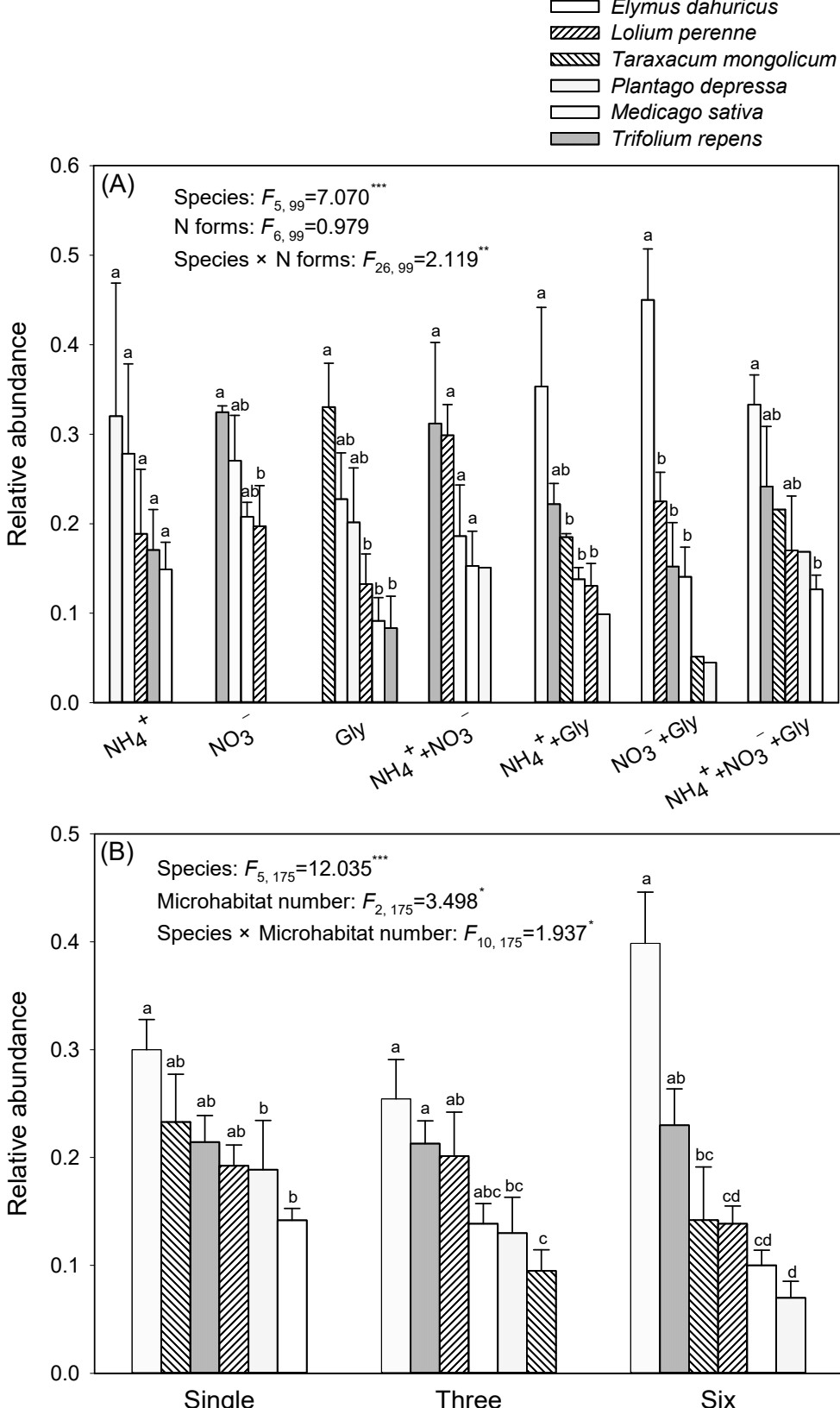

**Figure 4.** Relative abundance of the six species, ranked from the highest to the lowest, under the seven single-microhabitat treatments (**A**) and heterogeneous soils with different number of microhabitats (**B**). Different letters (a–d) at the end of bars indicate significant difference in relative abundance among the six species under each treatment. Symbols give * $p < 0.05$, ** $p < 0.01$ and *** $p < 0.001$.

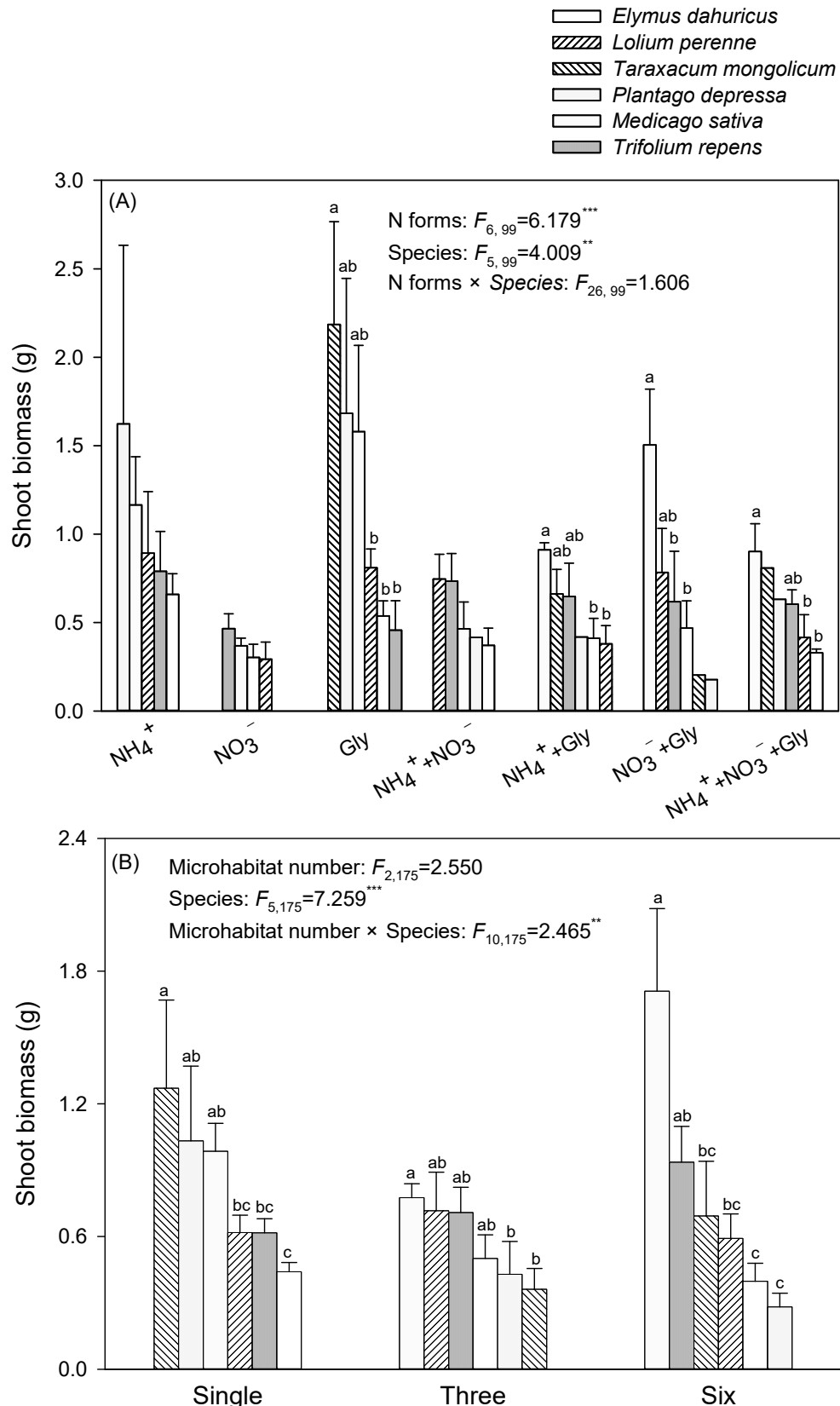

**Figure 5.** Shoot biomass of the six species, ranked from the highest to the lowest, under the seven single-microhabitat treatments (**A**) and heterogeneous soils with different number of microhabitats (**B**). Different letters (a–c) at the end of bars indicate significant difference in shoot biomass among the six species under each treatment. Symbols give ** $p < 0.01$ and *** $p < 0.001$.

## 4. Discussion

　　Our results showed a higher growth of plant community treated with organic N than plant community treated with other single types of solutions, probably due to an overall greater growth of the component species within the community. However, the nitrogen solution type did not influence the community evenness. Increasing the number of solution types did not alter the growth or evenness of the community, but it increased the difference in the relative abundance of the component species within the community. These results indicate that the type of nitrogen solution, i.e., the nitrogen form, may be more important than microhabitat diversity, in regulating community growth, and that the response at the species-level may not be cascaded to influence the response at the community-level.

　　We found that plant communities achieved the highest productivity when treated with organic N (i.e., glycine-N), compared to plant communities treated with other forms of nitrogen (i.e., ammonium and nitrate) or a combination of two or three forms of nitrogen. In general, plants prefer nitrate and ammonium to organic nitrogen, as the chemical nitrogen diffuses more rapidly, and can be acquired more easily by plant roots [25,31,32]. However, plants may utilize organic nitrogen compounds directly, especially when the nitrogen is limited [22,33–35]. In our study, the soil was nutrient-poor; thus, the plants—especially *Taraxacum mongolicum*—may have largely used the organic nitrogen, i.e., glycine, contributing to the high community productivity. This explanation was also confirmed by the positive relationship between the difference in competitive ability and plant community growth. Moreover, soil microbes may have stronger associations with glycine than with ammonium and nitrate e.g., [27,33,36–38]. We do not know how nitrogen-induced plant–soil microbe interactions may have been involved in the enhancement of plant community growth; despite this, our results indicate that organic nitrogen may have important implications in the maintenance of plant community productivity, and hence in stable ecosystem functions and services, especially when the ecosystem is nitrogen-poor [33,39].

　　Soil heterogeneity is generally thought to contribute to plant community growth and diversity, due to more microhabitats being available in the heterogeneous environment than in the homogeneous environment [11,40,41]. We expected that increasing microhabitat diversity would lead to increased plant productivity and diversity. Our results, however, did not support this idea. One possible explanation is that the size of the microhabitat (i.e., patch) in our experiment was smaller than the size of the plant rooting systems; therefore, the plants were able to utilize resources across microhabitats, leading to a reduced heterogeneity effect [11,42,43]. Alternatively, plants differ in their ability to absorb different chemical forms of nitrogen, and show species-specific preferential uptake of nitrogen forms [44–47]. Although the multi-microhabitats were created by adding different types of nitrogen solutions in a patchy manner, they all contained three nitrogen forms (i.e., ammonium, nitrate and glycine). This may have enabled the plants exhibiting high plasticity to shift their preferences in the utilization of different forms of nitrogen, depending on the realized environment [27]. These results, therefore, suggest that nitrogen forms (as discussed above), rather than the spatial distribution of the different forms of nitrogen, may be more important in influencing plant community performance.

　　Although the spatial heterogeneities of different nitrogen types had rather limited influences on the community properties, they had remarkable influences on the performance of plant species within the community. We found that plant growth was more unequal in heterogeneous soils containing a greater number of microhabitats, and that this growth difference did not eventually influence community evenness. This was in contrast to a previous study, which showed that soil heterogeneity consisting of low- and high-nutrient soil patches can promote plant community evenness by reducing the growth-difference of the component species [14]. The contrasting results were likely due to different manipulations of the soil heterogeneity. We manipulated the spatial heterogeneity of different nitrogen forms or combinations, and the total nitrogen in each microhabitat was equal. Therefore, plants exhibiting plasticity in utilization of nitrogen were able to dominate the community, and the subordinate species were unable to escape from competitive stress from

the dominant species, as they usually do in heterogeneous soils consisting of microhabitats of different qualities [48].

We conclude that plant community productivity responds to different forms or combinations of nitrogen, via changing competitive hierarchies between plant species within the community. Increasing microhabitat diversity, in terms of patches of different forms or combinations of nitrogen, increases plant growth difference, but this effect does not lead to changing productivity or diversity in the community. In contrast to previous studies, e.g., manipulating soil heterogeneity by varying the nitrogen availability of microhabitats [11,49], we manipulated soil heterogeneity by varying the type of nitrogen forms, but the nitrogen availability was consistent in different microhabitats. Our results add new empirical evidence to the traditional heterogeneity studies, that heterogeneous distribution of nitrogen forms may alter plant competitive hierarchies, which may have long-term influence on community structures.

**Author Contributions:** Conceptualization, B.H., W.X. and F.-H.Y.; methodology, B.H. and W.X.; validation, W.X., Z.-W.X. and F.-H.Y.; formal analysis, B.H.; investigation, B.H.; data curation, B.H.; writing—original draft preparation, B.H.; writing—review and editing, W.X., F.-H.Y. and Z.-W.X.; visualization, B.H. and W.X.; supervision, Z.-W.X. and W.X.; project administration, F.-H.Y. All authors have read and agreed to the published version of the manuscript.

**Funding:** This work was supported by the Central Government Guides the Local Science and Technology Development Foundation (2020ZY0027), National Natural Science Foundation of China (grant 32001122, 317611123001 and 32060284), the Natural Science Foundation of Inner Mongolia, China (2019JQ04) and Zhejiang Provincial Natural Science Foundation (grant LQ21C030003).

**Institutional Review Board Statement:** Not applicable.

**Data Availability Statement:** The data presented in this study are available on request from the corresponding author.

**Conflicts of Interest:** The authors declare no conflict of interest.

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
