# Peer review of "Species-Level Versus Community-Level Responses to Microhabitat Type and Diversity in an Experimental Plant Community"

_diversity, doi:10.3390/d14100803_

Round 1

Reviewer 1 Report

Row 34, and 47 correct "litter" to "little"

Row 131, insert "dry" before word "biomass"

Row 196, 201, use full name of the plant species Medicago sativa

Figure 4, Figure 5, suggestion to use full scientific names for all plant species used in the experiment. e.g. E. dahuricus correct to Elymus dahuricus.

Row 235, use full name, correct "T. mongolicum" to "Taraxacum mongolicum"

Author Response

Response to Reviewer 1 Comments

Row 34 and 47, correct "litter" to "little"

Response: Corrected. Thanks (Row 35, and 48).

Row 131, insert "dry" before word "biomass"

Response: Added as suggested (Row 132).

Row 196, 201, use full name of the plant species Medicago sativa

Response: As suggested, we used full name of the plant species Medicago sativa in the revised manuscript. (Row 199 and 205).

Figure 4, Figure 5, suggestion to use full scientific names for all plant species used in the experiment. e.g. E. dahuricus correct to Elymus dahuricus.

Response: As suggested, we used full scientific names for all plant species used in the experiments in Figure 4 and Figure 5. (Row 211 and 217)

Row 235, use full name, correct "T. mongolicum" to "Taraxacum mongolicum"

Response: Revised (Row 240).

Reviewer 2 Report

Dear Authors,

The introduction is presented correctly, in accordance with the subject. Numerous scientific articles, in concordance to the topic of the study, were consulted.

Methodology of the study was clearly presented, and appropriate to the proposed objectives.

The article has an interesting approach, and the results are presented correctly, in relation to the purpose of the study.

The discussions are appropriate, in the context of the results, and was conducted compared to other studies in the field.

The scientific literature, to which the reporting was made, is recent and representative in the field.

Some suggestions and corrections were made in the article.

The following aspects are brought to the attention of the authors.

1

To write the ionic forms correctly, it is recommended to use the Equation Editor in Word

It is “NH+” and not “4+”, respectively “NO-“ and not “3-

2.

Some portions of the text are highlighted with a gray background

eg

Page 1, row 27 to row 35

3.

It is recommended to check the way of writing for some values and units of measure

eg

“23.5 °C” with space (row 124)

“70°C” without space (row 130)

4.

It is recommended to check that the writing of some values is correct, relative to the English language

Eg

Page 5, Figure 2

“F6,14” or “F6.14

Page 6, Figure 3

“F1,33” or “F1.33

Page 7, Figure 4

“F5,99” or “F5.99

Page 8, Figure 5

“F6,99” or “F6.99

It is recommended to check for all F values

Author Response

Response to Reviewer 2 Comments

The introduction is presented correctly, in accordance with the subject. Numerous scientific articles, in concordance to the topic of the study, were consulted.

Methodology of the study was clearly presented, and appropriate to the proposed objectives.

The article has an interesting approach, and the results are presented correctly, in relation to the purpose of the study.

The discussions are appropriate, in the context of the results, and was conducted compared to other studies in the field.

The scientific literature, to which the reporting was made, is recent and representative in the field.

Some suggestions and corrections were made in the article.

The following aspects are brought to the attention of the authors.

Response: Many thanks.

1 To write the ionic forms correctly, it is recommended to use the Equation Editor in Word. It is “” and not “4+”, respectively “” and not “3-”.

Response: We used the Equation Editor and corrected these errors (Row 15 and 61).

2 Some portions of the text are highlighted with a gray background.

Eg, Page 1, row 27 to row 35

Response: All background has been removed in the revised manuscript (Row 27 to row 35).

3 It is recommended to check the way of writing for some values and units of measure.

Eg, “23.5 °C” with space (row 124)

“70°C” without space (row 130)

Response: Corrected (Row 124). We also double checked our manuscript and made corrections when necessary.

4 It is recommended to check that the writing of some values is correct, relative to the English language

Eg, Page 5, Figure 2, “F6,14” or “F6.14

Page 6, Figure 3, “F1,33” or “F1.33

Page 7, Figure 4, “F5,99” or “F5.99

Page 8, Figure 5, “F6,99” or “F6.99

It is recommended to check for all F values.

Response: We've double checked all F-values. We changed “F6,14” to “F6, 14”, similar changes are made for all F-values (Row 179, 191, 211 and 217).

Row 36, delete “but see”

Response: Deleted (Row 36).

Row 112, “Figure 1” instead of current from

Row 168, “(Figure 2A-B)”

Row 173, “(Figure 2A-B)”

Row 174, “(Figure 2C)”

Row 176, “(Figure 2D-F)”

Row 184, “(Figure 3A)”

Row 185, “(Figure 3C)”

Row 187, “(Figure 3B)”

Row 194, “(Figure 4A)”

Row 195, “(Figure 4A)”

Row 200, “(Figure 4A)”

Row 201, “(Figure 4B)”

Row 204, “(Figure 4B)”

Row 206, “(Figure 5)”

Response: Revised (Row 113, 169, 174, 175, 177,186, 187, 189, 197, 198, 203, 204, 207 and 209).

Row 161, “[]” reference number in addition

Response: Reference number added (Row 162).

Row 297, “,” instead of “:”

Response: Corrected, thanks (Row 302).
